# Federated $\mathcal{X}$-armed Bandit with Flexible Personalisation

**Ali Arabzadeh**                                    *s.arabzadeh@lancaster.ac.uk*
*School of Mathematical Sciences*
*Lancaster University*

**James A. Grant**                                    *j.grant@lancaster.ac.uk*
*School of Mathematical Sciences*
*Lancaster University*

**David S. Leslie**                                    *d.leslie@lancaster.ac.uk*
*School of Mathematical Sciences*
*Lancaster University*

**Reviewed on OpenReview:** *https://openreview.net/forum?id=fr61ToAd9o*

## Abstract

This paper introduces a novel approach to personalised federated learning within the $\mathcal{X}$-armed bandit framework, addressing the challenge of optimising both local and global objectives in a highly heterogeneous environment. Our method employs a surrogate objective function that combines individual client preferences with aggregated global knowledge, allowing for a flexible trade-off between personalisation and collective learning. We propose a phase-based elimination algorithm that achieves sublinear regret with logarithmic communication overhead, making it well-suited for federated settings. Theoretical analysis and empirical evaluations demonstrate the effectiveness of our approach compared to existing methods. Potential applications of this work span various domains, including healthcare, smart home devices, and e-commerce, where balancing personalisation with global insights is crucial.

## 1 Introduction

In the rapidly evolving field of machine learning, realising appropriate levels of personalisation has emerged as a critical challenge. This article addresses this pressing concern within the context of sequential decision-making, specifically focusing on the $\mathcal{X}$-armed bandit (Bubeck et al., 2011). The need for both data aggregation and individual customisation is evident in various domains. For instance, in the context of healthcare, data can be aggregated across subjects to yield population-level insights into (e.g., dosage efficacy), but individual differences necessitate personalised treatment. Similarly, for smart home devices, data aggregation across the clients can result in improved general policies, but each individual home has different characteristics; privacy should be guaranteed to encourage the beneficial aggregation of data for the common good, but personalised policies need to be implemented at individual devices. These scenarios highlight two key concepts: "federated learning" (Konečnỳ et al., 2016; McMahan et al., 2017), which enables data aggregation while maintaining privacy, and "personalisation"(Smith et al., 2017), which tailors strategies to individual needs. However, the optimal level of personalisation varies across different applications and use-cases; therefore, a one-size-fits-all approach is insufficient. This article develops a personalised federated learning framework for the $\mathcal{X}$-armed bandit, designed to allow flexible trade-offs between global and individual objectives, enhancing user experience by sharing exploration of the decision space, aggregating data securely, and facilitating individually beneficial decisions.

Federated learning (FL) (Konečnỳ et al., 2016; McMahan et al., 2017) is a distributed learning paradigm that addresses efficiency and privacy concerns in real-world, unbalanced and non-IID datasets. As datasets grow and models become more complex, the need to distribute the learning process across multiple machines

has increased. Conventional distributed learning approaches (Ma et al., 2017; Reddi et al., 2016; Zhang & Lin, 2015; Shamir et al., 2014) were designed for well-regulated environments like data centers, where data is typically balanced and i.i.d. across machines. However, these methods are ill suited for privacy-sensitive, heterogeneous data found on edge devices such as smartphones.

FL emerged as a solution to this challenge, enabling the training of models on rich and privacy-sensitive data from edge clients. In the FL framework, edge clients collaboratively train a shared global model under the coordination of a central server. This approach preserves the privacy of local data while reducing communication costs. By leveraging the collective knowledge of edge clients, FL offers a powerful new paradigm for distributed learning in real-world settings.

While vanilla FL has shown promise in privacy-preserving distributed learning, it can lead to suboptimal performance for individual edge clients with heterogeneous local datasets. To address this limitation, FL with personalisation has been proposed as an extension of the vanilla FL framework. This approach incorporates client-specific objectives to enhance model performance on heterogeneous data distributions(Smith et al., 2017; Wu et al., 2020).

In personalised FL, the globally trained model serves as a foundation that is further adapted or fine-tuned to better align with individual clients' specific data or preferences. This methodology enables the development of models that are both privacy-preserving and highly tailored to individual clients, effectively balancing collective knowledge sharing with local specialisation.

### 1.1 FL and Multi-Armed Bandits

While state-of-the-art FL research has predominantly focused on supervised learning scenarios, there is a growing interest in extending FL to the multi-armed bandit (MAB) framework (Lai & Robbins, 1985; Auer et al., 2002). The MAB problem is a popular framework for studying decision-making under uncertainty. In its simplest form, the MAB consists of a set of independent "arms", each providing a random reward from its corresponding probability distribution. An agent, without prior knowledge of these distributions, selects one arm in each round, aiming to maximise the total expected reward over time. This process necessitates a delicate balance between exploring arms to learn the unknown distributions and exploiting the current knowledge by choosing the arm that has historically provided the highest reward. The extension of FL to MAB is particularly relevant for applications like recommender systems and clinical trials, where decision-making is inherently distributed across multiple clients.

Unlike centralised MAB models that assume instant data access, the FL approach processes data locally at each client, thereby reducing communication overhead and enhancing data privacy. However, federated bandits introduce unique challenges beyond the classical exploration-exploitation trade-off in *centralised* models. These challenges include data heterogeneity and privacy preservation(Shi & Shen, 2021; Shi et al., 2021; Huang et al., 2021; Li et al., 2024b; Réda et al., 2022). In FL scenarios, collaboration among clients becomes crucial for accurate inference on the global model, especially given that each client's data may not be independent and identically distributed. Yet privacy concerns and communication costs motivate clients to avoid direct transmissions of local data.

The $\mathcal{X}$-armed bandit (XAB) problem is an extension of the classic MAB problem, where the decision-space is significantly larger and even continuous, involving a potentially infinite number of arms. Unlike traditional MAB problems which are limited to a discrete set of options, the XAB framework allows for a broader range of actions, making it particularly suited for complex environments where actions cannot be easily enumerated. This complexity introduces unique challenges in terms of exploration and exploitation, as the agent must efficiently navigate a vastly expanded action space to optimise outcomes. (Bubeck et al., 2011; Kleinberg et al., 2008)

Balancing exploration and exploitation while achieving privacy and personalisation requires a complex yet efficient communication protocol between clients and a central server. While Shi et al. (2021) have addressed this problem for finite, un-structured action spaces in their work on Personalised Federated MAB (PF-MAB), our work extends the principles of personalised federated learning to the XAB problem. In PF-MAB, the high probability guarantee that each client's active set eventually converges to a single best arm enables a

straightforward shift to pure local exploitation after the best arm is identified. This behavior is critical for ensuring minimal exploration once the optimal solution is reached. However, in the continuous action space of XAB, no such discrete "best arm" exists. Instead, the algorithm must continuously balance exploration and exploitation to approximate the optimal action effectively, as convergence to a single point does not apply in continuous settings. This necessitates a more refined exploration-exploitation trade-off to keep the algorithm both efficient and analytically tractable.

To effectively manage exploration in a continuous setting, our approach uses a partitioning strategy for the action space. This partitioning allows the algorithm to iteratively focus on progressively refined regions with high potential rewards. By adaptively adjusting the granularity of the partitions based on number of samples taken, the algorithm can efficiently explore and narrow down the action space without needing a singular "best" action to emerge.

Our research considers the critical addition of personalisation to recent works in federated XABs, particularly the novel method proposed by Li et al. (2024b). We realise this through a surrogate reward function that combines local and global knowledge, inspired by Hanzely & Richtárik (2020). Specifically, we defined a surrogate reward function for each client as a linear combination of their local reward function and the average reward function across all clients. This formulation inherently emphasises the importance of integrating global information, which is particularly beneficial in scenarios where data heterogeneity is significant and global insights are crucial for informed decision-making. Our approach offers a tunable balance between global and individualised optimisation, enabling clients to benefit from collective learning while still tailoring the model to their specific conditions.

While a recent extension (Li et al., 2024a) also considers a personalised variant of federated XAB, our approaches differ substantively. The approach of Li et al. (2024a) seeks to optimise each local functions directly, rather than personalised mixtures of local and global effects (as we do), under an assumption that differences among individuals are bounded (via constraints on local reward functions). This model and assumptions leads to an alternative notion of regret, for which the theoretical guarantees are ostensibly sharper, but since the settings are different, the results are not directly comparable. Our framework can capture broader heterogeneity among clients, and allows improved flexibility since the level of personalisation can be determined by the practitioner rather than environment alone.

The flexibility of our framework offers significant potential benefits across various domains. In healthcare applications, specifically clinical trials, this approach allows the derivation of general medical insights from aggregated data while keeping treatment recommendations personalized to each patient's medical history and condition. To better illustrate this, imagine several hospitals collaborating to determine the optimal dosage of a new drug. The action space, denoted as $\mathcal{X}$, represents the continuous range of possible dosages. Each hospital acts as a client, with a local objective function $\mu_m(x)$ that quantifies the effectiveness of dosage $x \in \mathcal{X}$ for their specific patient population. Factors such as demographics, medical history, and prevalent disease strains contribute to the uniqueness of each $\mu_m(x)$. The global model, aiming to ascertain the average drug effectiveness across all hospitals, utilizes the average of all local objective functions, represented as $\mu(x)$. The personalized objective function, denoted as $\mu'_m(x)$, combines each hospital's local objective with the global objective, striking a balance between individual hospital needs and overall drug efficacy.

## 1.2 Main Contributions

Our primary contributions are as follows:

1. We propose a novel personalised federated XAB model that accounts for user preferences and data heterogeneity. This model is strategically designed to optimise the trade-offs between communication efficiency and learning performance in a federated bandit setting.

2. We introduce `PF-XAB`, a novel algorithm tailored to the personalised federated XAB problem. Key features of `PF-XAB` include: a) edge clients communicate only their local reward estimates, preserving privacy of raw observations, b) the algorithm achieves sublinear regret while maintaining logarithmic communication cost.

3. We provide theoretical analysis demonstrating that `PF-XAB` achieves sublinear regret bounds while maintaining logarithmic communication cost.

## 2 Preliminaries

In this section, we define the framework for our study of the personalised federated XAB problem. We will outline our model and specific objectives, introducing the key notation and assumptions that underpin our analysis. For an integer $n \in \mathbb{N}$, we let $[n]$ represent the set of integers $\{1, 2, \ldots, n\}$. For a set $A$, $|A|$ denotes the size of set $A$.

### 2.1 Problem Formulation

**Clients and local models.** Let $\mathcal{X}$ be the measurable space of arms. We model the problem, following Li et al. (2024b), as a federation of $M \in \mathbb{N}$ clients, each with access to a distinct bounded *local objective* $\mu_m(x) : \mathcal{X} \to [0, 1]$, $m \in [M]$. These objectives may be non-convex, non-differentiable, and even discontinuous. As we focus on bounded functions, without loss of generality, we assume these functions are bounded within the unit interval $[0, 1]$.

Given a fixed number of rounds $T$, each client queries its local objective oracle once per round by selecting an arm $x_{m,t} \in \mathcal{X}$ in round $t \in [T]$. This evaluation yields a noisy feedback $r_{m,t} = \mu_m(x_{m,t}) + \epsilon_{m,t}$, where $\epsilon_{m,t}$ is a zero-mean, bounded random noise, independent of previous observations or other clients' observations. Importantly, the local objectives can capture client-specific preferences, i.e., $\mu_m(x)$ and $\mu_n(x)$ are not necessarily equal for distinct clients $m$ and $n$.

**The global model.** The global model is a $\mathcal{X}$-armed bandit model that shares the search space with the local models but uses the average of local objectives over all clients as its objective. This average, known as *global objective*, is defined as

$$\mu(x) := \frac{1}{M} \sum_{m=1}^{M} \mu_m(x), \quad x \in \mathcal{X}.$$

It is important to note that while the global model represents the average of local models, the global rewards are not directly accessible to any individual client, as local observations remain private.

**The personalised model.** In conventional federated learning framework, clients collaborate to optimise the global objective $\mu$. However, this approach may not always align with individual client interests. To address this, we capture the personalisation aspect via a mixed objective, inspired by the well-established and popular technique of Hanzely & Richtárik (2020) in federated learning literature.

Similar to (Shi et al., 2021; Salgia et al., 2023), we define a *personalised objective* $\mu'_m$, which is a convex combination of global and local objectives:

$$\mu'_m(x) := \alpha \mu_m(x) + (1 - \alpha)\mu(x), \quad x \in \mathcal{X}, \tag{1}$$

where $\alpha \in [0, 1]$ is the personalisation parameter.

The components of the mixed objective bear similarity to the objectives in (Li et al., 2024b) and (Li et al., 2024a), however handling the mixed objective brings additional challenges, as well as the flexibility to capture more diverse sets of clients. The personalisation parameter $\alpha$ allows for explicit control over the level of personalisation:

- $\alpha = 1$: Complete personalisation (equivalent to local objective)

- $\alpha = 0$: Complete reliance on the global model (no personalisation)

- $\alpha \in (0, 1)$: Balances global knowledge with local preferences

By varying $\alpha$, we address the trade-off between global and local performance. This model integrates personalisation while retaining the benefits of global generic knowledge, potentially mitigating overfitting risks common in fine-tuned models.

It is important to note that optimising this personalised objective is more challenging than scenarios where clients collectively optimise a generic global objective ($\alpha = 0$) or individually focus on local objectives ($\alpha = 1$). The mixed nature of the objective introduces complexities in balancing the local and global information.

In this personalised model, each client aims to optimise its own personalised objective, which is unique to that client. However, this introduces a significant challenge: the estimation of a client's personalised objective $\mu'_m$ requires knowledge of other clients' local objectives $\{\mu_n\}_{n \neq m}$, which are not directly observable by client $m$.

Optimising this personalised objective is more complex than traditional federated learning scenarios where clients either collectively optimize a global objective ($\alpha = 0$) or individually focus on local objectives ($\alpha = 1$). The mixed nature of $\mu'_m$ introduces complexities in balancing local and global information, particularly because client $m$ can only directly infer the local component of its personalised objective, while the global component, $(1 - \alpha)\mu(x)$, remains partially unobservable.

## 2.2 Performance Measure

In our proposed framework, we aim to devise a collaborative optimisation strategy across a federation of personalised models. We assess the performance of this strategy through the expected cumulative regret, which is essentially the sum of the individual clients' regrets, calculated with respect to their individual personalised objectives. Formally, we define the expected regret as

$$\mathbb{E}\left[\mathbb{R}(T)\right] = \mathbb{E}_\pi\left[\sum_{t=1}^{T}\sum_{m=1}^{M}\mu'^*_m - \sum_{t=1}^{T}\sum_{m=1}^{M}\mu'_m(x_{m,t})\right],$$

where $\mu'^*_m$ denotes the optimal value of $\mu'_m$ on $\mathcal{X}$. The expectation is taken with respect to randomness in the sequence of actions, which arises from the stochasticity of local rewards $r_{m,t}$.

It is important to note that in the context of regret, an inevitable increase over time is expected. Our focus, therefore, lies in understanding and minimising the rate at which this increase occurs, aiming to minimise the average regret over time.

## 2.3 Hierarchical Partitioning of Search Space

We employ a recursive partitioning approach denoted by $\mathcal{P} := \{\mathcal{P}_{h,i}\}_{h,i}$, which splits $\mathcal{X}$ into a hierarchy of nodes at various depths. The partitioning follows the rule:

$$\mathcal{P}_{0,1} = \mathcal{X}, \ \mathcal{P}_{h,i} := \mathcal{P}_{h+1,2i-1}\bigcup\mathcal{P}_{h+1,2i},$$

where:

- Each node $\mathcal{P}_{h,i}$ is defined by its depth $h$ and index $i$, corresponding to a specific region within the discretisation of the search space.

- For each $h \geq 0, \ i > 0$ the set $\{\mathcal{P}_{h+1,2i-j}\}_{j=0}^{1}$ consists of two disjoint children nodes of $\mathcal{P}_{h,i}$.

- At each depth $h$, the union of all nodes is equivalent to the entire space $\mathcal{X}$.

The partition is fixed and shared among all clients and the central server prior to the beginning of the FL process. This ensures a consistent structure for exploration and information sharing across the federation. While we employ binary partitioning of the space in this work, our principles could readily be adapted to a $k$-ary partitioning approach.

This hierarchical partitioning provides a structured approach to exploring the continuous action space $\mathcal{X}$, facilitating efficient search and information sharing in our federated XAB framework.

### 2.4 Assumptions

To evaluate the performance of the proposed algorithms, we employ a set of assumptions commonly used in prior works on $\mathcal{X}$-armed bandit (Bubeck et al., 2011; Gheshlaghi Azar et al., 2014; Grill et al., 2015; Li et al., 2024b).

**Assumption 2.1. (Dissimilarity Function)** The space $\mathcal{X}$ is equipped with a dissimilarity function $\ell : \mathcal{X}^2 \to \mathbb{R}$ such that $\ell(x,y) \geq 0, \forall(x,y) \in \mathcal{X}^2$ and $\ell(x,x) = 0$.

Given a dissimilarity $\ell$, we define:

- Diameter of any subset $A \subseteq \mathcal{X}$: $\mathrm{diam}(A) := \sup_{x,y \in A} \ell(x,y)$

- Open ball with a radius $r$ centered at $c$: $\mathcal{B}(c,r) := \{x \in \mathcal{X} : \ell(x,c) \leq r\}$

**Assumption 2.2. (Local Smoothness)** We assume that there exist $\nu_1, \ \nu_2 > 0$ and $0 < \rho < 1$ such that for all nodes $\mathcal{P}_{h,i}, \mathcal{P}_{h,j} \in \mathcal{P}$ on depth $h$,

- $diam(\mathcal{P}_{h,i}) \leq \nu_1 \rho^h$

- $\exists \ x_{h,i} \in \mathcal{P}_{h,i}$ s.t. $\mathcal{B}_{h,i} := \mathcal{B}(x_{h,i}, \nu_2 \rho^h) \subset \mathcal{P}_{h,i}$

- $\mathcal{B}_{h,i} \bigcap \mathcal{B}_{h,j} = \emptyset$ for all $i \neq j$

- For any objective $\mu \in \{\mu_1, \dots, \mu_m\}$ we have for all $x, y \in \mathcal{X}$,

$$\mu^* - \mu(y) \leq \mu^* - \mu(x) + \max\{\mu^* - \mu(x), \ell(x,y)\}.$$

This assumption pertains to the "smoothness" of the objective functions and ensures that as the depth increases, the search regions become increasingly fine, allowing for a more detailed exploration of promising areas within the hierarchical partitioning approach.

*Remark* 2.3. Similar to the existing works on the $\mathcal{X}$-armed bandit problem, knowledge of dissimilarity function $\ell$ is not required for our algorithm. However, knowledge of the smoothness constants $\nu_1, \ \rho$ is necessary.

*Remark* 2.4. In this setup, we do not impose specific assumptions on clients' objective functions beyond the regular smoothness conditions typical in $\mathcal{X}$-armed bandit literature. This contrasts with (Li et al., 2024a), the only prior research on this problem and most related and similar to ours, which imposes two additional constraining assumptions (Assumptions 3 and 4 in their work). These assumptions jointly require that near-optimal points with respect to the global objective are also near-optimal with respect to all local objectives. Our problem setup is more flexible, allowing for a wider range of real-world scenarios with varying degrees of data heterogeneity.

## 3 Algorithm and Analysis

In this section, we introduce a novel phase-based elimination algorithm to address the challenges of personalised federated XAB. We highlight its distinctive features compared to existing algorithms and provide a theoretical analysis of its performance.

A bespoke algorithm is sorely necessitated for the personalised federated XAB, as existing XAB algorithms will fail to meet the challenges of the setting. Unlike the focus on optimisation of the *global objective* as explored in works by Shi & Shen (2021) and Li et al. (2024b), our interest lies in simultaneously optimising the *personalised objectives* of the clients. This task is evidently more challenging since it can be viewed as a problem of multi-objective optimisation with communication budget constraint.

Consequently, developing an effective communication strategy that enables an unbiased estimation of the global objective becomes a critical element of algorithm design. This is particularly important in scenarios involving a large number of clients. Furthermore, the constraints on communication budget mean that local rewards are not instantly accessible. As a result, algorithms that depend on immediate reward feedback, such

as `HOO`(Bubeck et al., 2011) and `HCT` (Gheshlaghi Azar et al., 2014), are not appropriate for addressing this problem.

In the context of federated bandit algorithms, the requirement for minimal communication overhead indeed renders algorithms that are built on phase-based elimination techniques appealing(Shi & Shen, 2021; Shi et al., 2021; Réda et al., 2022). Therefore, algorithms such as our own proposal, which eliminate sub-optimal nodes in phases, are more suitable for the federated XAB problem, as suggested in works like (Li et al., 2024b;a).

### 3.1 The `PF-XAB` Algorithm

We now introduce our new phased-elimination algorithm, called Personalised Federated $\mathcal{X}$-Armed Bandit (`PF-XAB`), designed to address the challenges outlined above.

---

**Algorithm 1** Fed-XAB: server

---

**Input:** $T$, $M$
**while** not reaching the time horizon $T$ **do**
    Receive $\mathcal{A}^m(p)$ from all clients
    Broadcast $\mathcal{A}(p) = \bigcup_{m \in [M]} \mathcal{A}^m(p)$ and $f(p) = \frac{2\log(T)}{M\nu_1^2\rho^{2h}}$
    Receive local estimates $\{\bar{\mu}_{(h,i),m}\}_{m \in [M],(h,i) \in \mathcal{A}^m(p)}$
    **for** every $(h, i) \in \mathcal{A}(p)$ **do**
        Update $\bar{\mu}_{(h,i)}(p) = \frac{1}{M} \sum_{m=1}^{M} \bar{\mu}_{(h,i),m}(p)$
    **end for**
    Broadcast $\{\bar{\mu}_{(h,i)}(p)\}_{(h,i) \in \mathcal{A}(p)}$
    $p = p + 1$
**end while**

---

The `PF-XAB` algorithm comprises two components: a client-side algorithm (Algorithm 2) and a server-side algorithm (Algorithm 1). The algorithm operates in dynamic phases, leveraging the hierarchical partition to gradually identify the optimum by systematically eliminating sub-optimal nodes within the search domain.

The server acts as a coordinator with two primary functions. It broadcasts active nodes $\mathcal{A}(p)$ in phase $p$, which represent the collective active nodes across all clients. Additionally, It aggregates empirical estimates of the local objectives $\bar{\mu}_{(h,i)}(p)$ for all active nodes $(h, i) \in \mathcal{A}(p)$ from clients, and subsequently broadcasts the empirical estimates of the global objective back to all clients.

On the client side, the algorithm involves a phased interaction with the environment, segmented into three distinct sub-phases: global exploration, local exploration, and exploitation. During both the global and local exploration sub-phases, clients take actions from two sets: the global active set $\mathcal{A}(p)$, and their own local active set $\mathcal{A}^m(p)$. They use the feedback to these actions to construct empirical estimates of their local objective. These local estimates are then communicated to the server, allowing for centralized estimation of global objective. While a client waits for the server to provide the empirical estimates of the global objective (as other clients may have more exploration to perform), it enters the exploitation sub-phase, where it simply exploits the best action based on the most recent evaluations.

`PF-XAB` offers a degree of privacy and confidentiality by sharing only average rewards $(\bar{\mu}_{(h,i),m}(p))$ with the server and not individual reward values $(r_{m,(h,i),t})$.The server then shares summary statistics with other clients. However, we acknowledge the need for a more formal treatment of privacy considerations in this setting.

---

**Algorithm 2** PF-XAB: $m$-th client

---

**Input:** partitioning $\mathcal{P}$, time horizon $T$, client count $M$, personalization parameter $\alpha$, stopping depth $H$
**Initialize** $p = 1$, $\mathcal{A}^m(1) = \{(1,1),(1,2)\}$
**while** not reaching the depth $H$ **do**
    Receive $\mathcal{A}(p)$ and $f(p)$ from the server
    **for** each $(h,i) \in \mathcal{A}(p)$ sequentially **do**                  ▷ Global exploration
        Pull the node $\lceil(1-\alpha)f(p)\rceil$ times and receive rewards $\{r_{m,(h,i),t}\}$
        Update $s_{(h,i),m} = s_{(h,i),m} + \sum_t r_{m,(h,i),t}$ and $T_{m,(h,i)} = T_{m,(h,i)} + \lceil(1-\alpha)f(p)\rceil$
    **end for**
    **for** each $(h,i) \in \mathcal{A}^m(p)$ sequentially **do**               ▷ Local exploration
        Pull the node $\lceil M\alpha f(p)\rceil$ times and receive rewards $\{r_{m,(h,i),t}\}$
        Update $s_{(h,i),m} = s_{(h,i),m} + \sum_t r_{m,(h,i),t}$ and $T_{m,(h,i)} = T_{m,(h,i)} + \lceil M\alpha f(p)\rceil$
    **end for**
    Calculate $\bar{\mu}_{(h,i),m}(p) = s_{(h,i),m}/T_{m,(h,i)}$ for every $(h,i) \in \mathcal{A}(p)$
    Send $\{\bar{\mu}_{(h,i),m}(p)\}_{(h,i)\in\mathcal{A}(p)}$ to the server
    Set $(h_p, i_p) = \arg\max_{(h,i)\in\mathcal{A}^m(p)}\{\bar{\mu}_{(h,i),m}(p)\}$
    Pull node $(h_p, i_p)$ until $\{\bar{\mu}_{(h,i)}(p)\}_{(h,i)\in\mathcal{A}(p)}$ are received from the server      ▷ Exploitation
    **for all** $(h,i) \in \mathcal{A}^m(p)$ **do**
        $\bar{\mu}'_{(h,i),m}(p) = \alpha\bar{\mu}_{(h,i),m}(p) + (1-\alpha)\bar{\mu}_{(h,i)}(p)$
    **end for**
    Compute the elimination set $\mathcal{E}^m(p) = \left\{(h,i) \in \mathcal{A}^m(p) \mid \bar{\mu}'_{(h,i),m}(p) + \nu_1\rho^h \le \bar{\mu}'_{(h_p,i_p),m}(p) - 2B_p\right\}$
    Compute the new set of active nodes

$$\mathcal{A}^m(p+1) = \{(h+1, 2i-j) \mid j \in \{0,1\}, (h,i) \in (\mathcal{A}^m(p) \setminus \mathcal{E}^m(p))\}$$

    Send $\mathcal{A}^m(p+1)$ to the server
    $p = p + 1$
**end while**
Select and play the optimal action for the remainder of the time

---

The algorithm details are presented in Algorithms 1 and 2. In these algorithms, we use the notation $(h,i)$ to index node $\mathcal{P}_{h,i}$ in the partitioning tree. $T_{m,(h,i)}$ denotes the number of pulled samples from $m$-th local objective in $(h,i)$. By convention, we always sample the midpoint when playing a node. Due to the structure of the algorithm, $T_{m,(h,i)}$ is equal to $\lceil M\alpha f(p)\rceil + \lceil(1-\alpha)f(p)\rceil$ and $\lceil(1-\alpha)f(p)\rceil$ for local and global active nodes, respectively. Consequently, the total number of samples taken to compute the empirical mean of the personalised objective $\bar{\mu}'_{m,(h,i)}$ is lower bounded by $\lceil M\alpha f(p)\rceil + M\lceil(1-\alpha)f(p)\rceil \ge Mf(p)$ and the corresponding confidence bound in elimination rule is $B_p = c\sqrt{\frac{\log(T)}{Mf(p)}}$.

### 3.2 Theoretical Analysis

This section contains our main theoretical analysis and results. Here, we establish an upper bound on the cumulative regret of the PF-XAB algorithm, operating under the assumptions previously outlined. Before this statement in Theorem 3.9, several key lemmas are introduced. Analysing the regret of PF-XAB is more complex than that of algorithms like Fed-PNE because PF-XAB does not directly observe the personalised objectives $\mu'(m)$. Instead, it works with estimates $\bar{\mu}_{(h,i),m}(p)$ constructed from local and global objective estimates. The proof must carefully account for the discrepancy between these estimates and the true personalised objectives.

We first determine the variance proxy of the empirical estimates of personalised objectives in Lemma 3.3, and use this to construct a high-probability "good event" in Lemma 3.4, under which the empirical estimates of personalised objectives are sufficiently close to the actual personalised objectives. Lemmas 3.5 and 3.6 demonstrate two key aspects: firstly, that the optimal nodes (those containing the optimal point) will never be eliminated, and secondly, that all points within the remaining region are, in fact, sub-optimal. Proofs of the lemmas are deferred to Appendix B.

To proceed we recall some facts concerning sub-Gaussian random variables.

**Definition 3.1. (Sub-Gaussian Random Variables)** A zero-mean random variable $X \in \mathbb{R}$ is said to be sub-Gaussian with variance proxy $\sigma^2$ if $\mathbb{E}[X] = 0$ and its moment generating function satisfies

$$\mathbb{E}[\exp(sX)] \leq \exp\left(\frac{\sigma^2 s^2}{2}\right), \ \forall s \in \mathbb{R}.$$

**Definition 3.2. (Sum of sub-Gaussian Random Variables)** If $\{X_1, X_2, \cdots, X_n\}$ are independent $\{\sigma_1^2, \sigma_2^2, \cdots, \sigma_n^2\}$-sub-Gaussian random variables, then $Y = \sum_{i=1}^{n} a_i X_i$, for any $a \in \mathbb{R}^n$, is a $\tilde{\sigma}^2(Y)$-sub-Gaussian random variable, where

$$\tilde{\sigma}^2(Y) = \sum_{i=1}^{n} a_i^2 \sigma_i^2.$$

As discussed, we first establish the sub-Gaussianity of our estimators, and from this build a high-probability event.

**Lemma 3.3. (Sub-Gaussian Estimators)** *Let $\bar{\mu}'_{(h,i),m}(p)$ denote the empirical estimate of $\mu'_{(h,i),m}$ at the end of phase $p$. Then, $\bar{\mu}'_{(h,i),m}(p)$ is a sub-Gaussian random variable with variance proxy $\sqrt{\frac{1}{4Mf(p)}}$.*

**Lemma 3.4. (High Probability Event)** *Define the* good event $G_T$ *as*

$$G_T := \left\{\forall p \in P_T, \forall m \in [M], \forall (h,i) \in \mathcal{A}^m(p) : \left|\bar{\mu}'_{(h,i),m}(p) - \mu'_{(h,i),m}\right| \leq c\sqrt{\frac{\log(T)}{Mf(p)}}\right\},$$

*where the right hand side is the confidence bound $B_p$ for node $\mathcal{P}_{h,i}$ and $c = \sqrt{2}$ is a constant. Then for any fixed $t$, we have $\mathcal{P}(G_T) \geq 1 - 2M^2 T^{-3}$*

On this high-probability event, we can establish guarantees both that the optimal node will not be eliminated, and that all other non-eliminated nodes are of a good quality.

**Lemma 3.5. (Permanence of Optimal Nodes)** *Under the assumption that the high probability event $G_T$ holds, it can be stated for any client $m$ that by the end of phase $p \in P_T$, the node $\mathcal{P}_{h_p, i^*_{m,p}}$ will not be eliminated. This particular node contains the global optimum $x^*_m$ of the personalised objective $\mu'_m(x)$ at the depth $h_p$. More precisely, this implies that $(h_p, i^*_{m,p})$ will not be included in the set of eliminating nodes $\mathcal{E}^m(p)$.*

**Lemma 3.6. (Quality of Un-eliminated Nodes)** *Assuming the high probability event $G_T$ holds and $f(p) = \frac{2\log(T)}{M\nu_1^2 \rho^{2h}}$, all points in every un-eliminated nodes upon the conclusion of phase $p$, is at least $12\nu_1 \rho^{h_p}$-optimal.*

The algorithm's performance is sensitive to the smoothness parameters $\nu_1$ and $\rho$, which influence the confidence bounds in the elimination process. A conservative estimate of $\rho$ results in smaller confidence bounds and shorter exploration phases, potentially leading to premature convergence to suboptimal regions and unbounded regret. In contrast, overestimating $\rho$ produces larger confidence bounds and longer exploration phases, yielding regret commensurate with this estimate. Therefore, accurate estimation of these parameters is crucial for achieving robust performance.

To complete the proof of Theorem 3.9, we need one additional concept: that of near-optimality dimension.

**Definition 3.7. (Near-optimality dimension)** Given a function $f : \mathcal{X} \to \mathbb{R}$, the $\nu$-*near-optimality dimension*, denoted by $d_f(\epsilon, \nu\epsilon)$, is the smallest $d > 0$ such that there exists $C > 0$ such that for any $\epsilon > 0$, the covering number of $\epsilon$-optimal subset of $\mathcal{X}$, denoted as $\mathcal{X}_{f,\epsilon}$, with $\ell$-balls of radius $\nu\epsilon$ is less than $C\epsilon^{-d}$.

*Remark* 3.8. The near-optimality dimension $d$ reflects the complexity of an objective function's landscape. For smooth functions with few local optima, $d$ is usually low, as near-optimal points remain limited even

with a larger search radius. In contrast, highly non-smooth functions with many local optima tend to have a higher $d$, complicating the search for the global optimum. Typically, $d$ is an intrinsic property of the problem and cannot be directly controlled. In many black-box optimisation cases, $d = 0$ is common, indicating a simpler landscape, as observed by Bubeck et al. (2011) and Valko et al. (2013).

Using the definition of near-optimality dimension, we denote $d_m = d_{\mu'_m}(12\nu_1\rho^h, \nu_1\rho^h)$ for every $m \in [M]$.

We now present the central result that establishes an upper bound on the expected cumulative regret of the `PF-XAB` algorithm.

**Theorem 3.9. (Regret Upper Bound)** *Suppose that $\mu_m(x)$ satisfies Assumptions 2.2, and let $d_m$ denote the near-optimality dimension of the mixed objective as defined in Definition 3.7. The expected cumulative regret of the `PF-XAB` algorithm is bounded above by*

$$\mathbb{E}[R(T)] = \widetilde{\mathcal{O}}(T^{\frac{d'+2}{d'+3}}),$$

*where $d' = \max\{d_1, \ldots, d_M\}$.*

*Proof.* Let $\mathbb{1}_G$ denote the indicator function associated with event $G$. Firstly we decompose the regret into two terms based on the presence of the good event

$$R(T) = \sum_{m=1}^{M} \sum_{t=1}^{T} \mu'^*_m - \mu'_{(h_t,i_t),m} = \sum_{m=1}^{M} \sum_{t=1}^{T} \left(\mu'^*_m - \mu'_{(h_t,i_t),m}\right) \mathbb{1}_{G_T} + \sum_{m=1}^{M} \sum_{t=1}^{T} \left(\mu'^*_m - \mu'_{(h_t,i_t),m}\right) \mathbb{1}_{G_T^c}.$$

Thus $\mathbb{E}\left[R(T)\right] = \mathbb{E}\left[R^G(T)\right] + \mathbb{E}\left[R^{G^c}(T)\right]$.

The second term of expected regret can be bounded as follows by using Lemma 3.4:

$$\mathbb{E}\left[R^{G^c}(T)\right] = \mathbb{E}\left[\sum_{m=1}^{M} \sum_{t=1}^{T} \left(\mu'^*_m - \mu'_{(h_t,i_t),m}\right) \mathbb{1}_{G_T^c}\right] \leq \sum_{m=1}^{M} \sum_{t=1}^{T} \mathcal{P}(G_T^c) \leq 2M^3 T^{-2}.$$

To bound $\mathbb{E}\left[R^G(T)\right]$, we first bound the regret that we incur under the good event within each phase. This regret could be decomposed into three parts, incurred by three sub-phases that we have in the proposed algorithm.

$$R^G(T) = \sum_{p \in P_T} R^p_{\text{G-expr}} + R^p_{\text{L-expr}} + R^p_{\text{expt}}$$

Here, $R^p_{\text{G-expr}}$, $R^p_{\text{L-expr}}$, and $R^p_{\text{expt}}$ denote the regret within phase $p$ that incurs to the algorithm in the global exploration, local exploration and exploitation sub-phases, respectively.

During the global exploration sub-phase, each client has to pull non-optimal arms from the active nodes of other clients included in $\mathcal{A}(p)$. Therefore, the regret term associated to the global exploration $R^p_{\text{G-expr}}$, can be upper bounded by the length of this sub-phase

$$\mathbb{E}^G[R^p_{\text{G-expr}}] \leq |\mathcal{A}(p)| \lceil (1-\alpha)f(p) \rceil \leq M \cdot \max_{m \in [M]} |\mathcal{A}^m(p)| \lceil (1-\alpha)f(p) \rceil.$$

The nodes in $\mathcal{A}^m(p)$ are direct descendants of the un-eliminated nodes in $\mathcal{A}^m(p-1) \setminus \mathcal{E}^m(p-1)$, and the size of $\mathcal{A}^m(p-1) \setminus \mathcal{E}^m(p-1)$ is bounded by $C\rho^{-d_m h_{p-1}}$ by the definition of near-optimality dimension (Definition 3.7) and Lemma 3.6. Therefore, it means

$$|\mathcal{A}^m(p)| = 2|\mathcal{A}^m(p-1) \setminus \mathcal{E}^m(p-1)| \leq 2C\rho^{-d_m h_{p-1}}$$

and results the following bound

$$\mathbb{E}^G[R^p_{\text{G-expr}}] \leq |\mathcal{A}(p)| \lceil (1-\alpha)f(p) \rceil \leq 2MC\rho^{-h_{p-1}d'} \times 2(1-\alpha)\frac{2\log(T)}{M\nu_1^2\rho^{2h_p}} \quad = \frac{8C(1-\alpha)\log(T)}{\nu_1^2\rho^2}\rho^{-h_{p-1}(d'+2)}.$$

Next, we bound the regret incurred in the rest of each phase, given by $R^p_{\text{L-expr}} + R^p_{\text{expt}}$. From Lemma 3.6, we know that all the actions clients take in these two sub-phases are sub-optimal, thus we can establish the following upper bound:

$$\mathbb{E}^G[R^p_{\text{L-expr}} + R^p_{\text{expt}}] \leq 12\nu_1\rho^{h_{p-1}} \times \lceil M\alpha f(p)\rceil \times \max_{m\in[M]}|\mathcal{A}^m(p)| \tag{2}$$

$$\leq 12\nu_1\rho^{h_{p-1}} \times 2M\alpha\frac{2\log(T)}{M\nu_1^2\rho^{2h_p}} \times 2C\rho^{-d'h_{p-1}} \leq \frac{96C\alpha\log(T)}{\nu_1\rho^2}\rho^{-h_{p-1}(d'+1)}. \tag{3}$$

Having established the key components of proof, we now proceed to the final step of the proof. It involves determining the depth $H$ at which we will halt exploration and start to do pure exploitation. The expected regret under the presence of the good event is bounded as follows

$$\mathbb{E}^G[R(T)] \leq \frac{8C(1-\alpha)\log(T)}{\nu_1^2\rho^2}\sum_{p=1}^P\left(\rho^{-(d'+2)}\right)^{h_{p-1}} + \frac{96C\alpha\log(T)}{\nu_1\rho^2}\sum_{p=1}^P\left(\rho^{-(d'+1)}\right)^{h_{p-1}} + 12\nu_1\rho^{h_P}(T-T_P)$$

$$\leq \frac{16Mc(1-\alpha)\log(T)}{\nu_1^2\rho^2(\rho^{-(d'+2)}-1)}\left(\rho^{-(d'+2)}\right)^H + \frac{96M^2\alpha c\log(T)}{\nu_1\rho^2(\rho^{-(d'+1)}-1)}\left(\rho^{-(d'+1)}\right)^H + 12\nu_1\rho^H T,$$

where $H$ represents the depth of the partitioning tree by the end of phase $P$. Thus, the proof concludes by putting everything together as

$$\mathbb{E}[R(T)] \leq 2M^3T^{-2} + \frac{16Mc(1-\alpha)\log(T)}{\nu_1^2\rho^2(\rho^{-(d'+2)}-1)}\left(\rho^{-(d'+2)}\right)^H + \frac{96M^2\alpha c\log(T)}{\nu_1\rho^2(\rho^{-(d'+1)}-1)}\left(\rho^{-(d'+1)}\right)^H + 12\nu_1\rho^H T, \tag{4}$$

and choosing $H$ such that $\rho^H = O(T^{-1/d'+3})$.

$\square$

While a bespoke lower bound for the personalised federated XAB problem is not yet available to assess the tightness of this bound, we can compare our $\widetilde{\mathcal{O}}(T^{\frac{d'+2}{d'+3}})$ to results for similar problems. Notably, based on the lower bound by Bubeck et al. (2011), the lower bound for the scenario in which all clients have complete access to each other's rewards at all timesteps is of order $\mathcal{O}(MT^{\frac{d'+1}{d'+2}})$ where $d'$ denotes the maximum near-optimality dimension among the personalised objectives. This bound, which holds under centralised conditions, provides a useful baseline for evaluating the performance of our approach in the federated setting. Providing a regret lower bound for the personalised federated XAB problem, whether under the definition of PF-PNE with respect to local objectives or our definition with respect to personalised objectives, remains an open problem.

The setting of Theorem 3.9 is more general compared to that of Li et al. (2024a), and therefore the slightly increased order of the regret bound is perhaps unsurprising. Under the restrictive assumptions of (Li et al., 2024a), our algorithm achieves the same regret upper bound as PF-PNE, which is of order $\widetilde{\mathcal{O}}(T^{\frac{d'+1}{d'+2}})$. A sketch of the proof for this claim is provided in the appendix. In any case, the existence of a method which is competitive under stronger assumptions and robust enough to attain sublinear regret under our weaker assumptions is encouraging with a view to real-world challenges where strong assumptions may be challenging to verify.

*Remark* 3.10 (Communication Cost). The number of communication **rounds** is always depends logarithmically on $T$, as the length of each phase $h$ is proportional to $\rho^{-(d'+1)h}$. The communication **cost** measured by the number of messages exchanged between clients and the server, can be bounded by $CM\log(T)T^{\frac{d+2}{d+3}}$, where $C$ is a constant.

*Remark* 3.11 (Effect of Personalisation Parameter). As shown in (4), the personalisation parameter $\alpha$ influences the second and third terms of the regret bound. However, the third term is dominated by the second and fourth terms, which masks the impact of $\alpha$ on the overall bound. We conjecture that this lack of dependence on $\alpha$ may be due to the upper bound not being tight, as well as the absence of a parameter bounding the differences between local objectives. Establishing such a bound could reduce the pessimism in our analysis, potentially clarifying the role of personalisation parameter in the regret performance.

## 4 Experiments

In this section we evaluate the empirical performance of `PF-XAB` through a series of experiments employing synthetic objective functions and real-world dataset. To provide a comprehensive assessment, we benchmark `PF-XAB` against relevant existing algorithms. Specifically, we compare to the federated $\mathcal{X}$-armed bandit algorithm `Fed-PNE` (Li et al., 2024b) and the personalised federated $\mathcal{X}$-armed bandit algorithm `PF-PNE` (Li et al., 2024a). Each of these methods adopts a different notion of regret, reflecting their distinct models, which makes a truly fair comparison challenging. However, we choose to compare to closest counterparts in the literature, not to suggest their perfect compatibility with our problem, but rather to illustrate the necessity of a bespoke solution. This comparison highlights a performance gap when the algorithms of (Li et al., 2024b) and (Li et al., 2024a) are applied to our specific problem domain. Throughout this section, all experiments has been conducted on a federation with $M = 10$ clients.

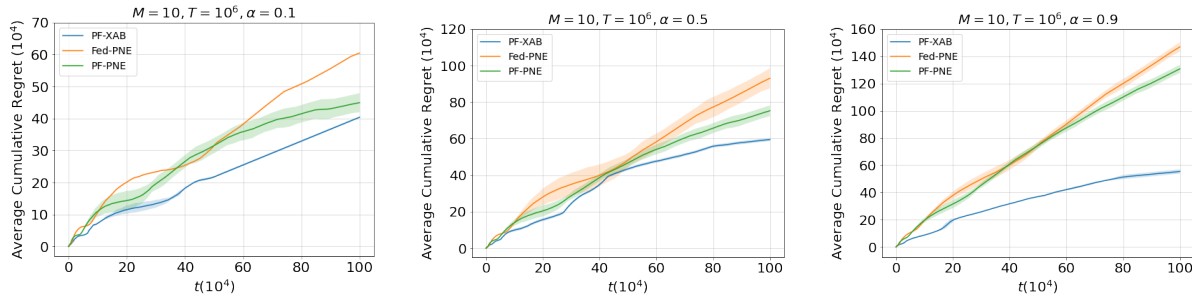

Figure 1: Comparison of Cumulative Regret for `PF-XAB`, `Fed-PNE`, and `PF-PNE` Algorithms on the Garland Dataset Across Varying Levels of Personalisation $\alpha$, with $M = 10$ machines over $T = 2 \times 10^6$ Time Steps. Each graph demonstrates the impact of personalisation on the algorithms' performance, with $\alpha$ representing the degree of personalisation from low ($\alpha = 0.1$) to high ($\alpha = 0.9$).

We first conduct a range of experiments on two synthetic objective functions: the Garland and DoubleSine functions with varying levels of personalisation in the regret definition and in `PF-XAB`. These two synthetic functions are frequently used in the experiments of $\mathcal{X}$-armed bandit algorithms due to their large number of local optima and extreme non-smoothness (Gheshlaghi Azar et al., 2014; Grill et al., 2015; Shang et al., 2019; Bartlett et al., 2019). The experimental design takes the canonical functions as a baseline and modulates these such that each client's local optimum is distinct, whereas the global optimum is strategically positioned to be sub-optimal with respective to the local objectives. The average cumulative regrets of different algorithms are provided in Figure 1 and 4 (see Appendix B) for the Garland and DoubleSine functions, respectively. The curves in these figures represent the averages over 10 independent runs of each algorithm, with the shaded regions indicating 1-standard deviation error bars.

In addition to our initial experiments with synthetic objective functions, we extended our experiments to a real-world-inspired scenario using the landmine dataset (Liu et al., 2007). This dataset comprises data from multiple landmine fields, each assigned to a client, with features extracted from radar images to determine the presence of landmines at each location. Each client aims to optimise their performance on a classification task, using a Support Vector Machine (SVM) classifier equipped with RBF kernels. Our experiments explore $d = 2$ kernel parameters, $\gamma$ ranging in $[0.01, 10]$, and the regularisation parameter $C$, selected from the range $[10^{-4}, 10^4]$, forms the domain space of our experiments. The local objective for each client is defined as the AUC-ROC score of the classifier on their assigned landmine dataset. The performance of different algorithms

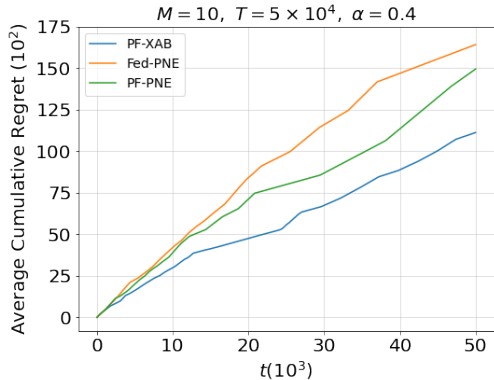

Figure 2: Cumulative regret comparison of `PF-XAB`, `Fed-PNE`, and `PF-PNE` on the Landmine dataset.

in this real-world setting is illustrated in Figure 2. This figure presents the average cumulative regret of each algorithm with $\alpha = 0.4$, showcasing our algorithm's effectiveness by its ability to achieve the smallest regret. This outcome not only validates our approach in synthetic settings but also demonstrates its practical applicability and superiority in handling complex, real-world tasks, such as landmine detection.

In the third and final experiment, our intention was to assess `PF-XAB`'s performance in achieving a balance between personalisation and generalisation across the federated landscape. Figure 3 indicates that `PF-XAB` successfully converges to optimal choices across different values of $\alpha$, effectively demonstrating its proficiency in balancing the trade-off between personalisation and generalisation. This figure outlines the averaged per-step reward attained by the `PF-XAB` under varying values of $\alpha \in [0, 1]$. It benchmarks these results against the theoretical maximums for both global and local mean rewards, referred to as "best global" and "best local", respectively.

The analysis encompasses three definition of rewards: personalised, global, and local, each labeled accordingly and derived by clients' actions. At $\alpha = 0$, both personalised and global rewards align with the optimal global reward, while local rewards are significantly sub-optimal. As $\alpha$ is progressively increased, the personalised and local rewards trend up, indicating a shift in focus towards optimizing local rewards, and simultaneously the global reward trends down. At $\alpha = 1$, both personalised and local rewards almost approximate the optimal local reward, while the global rewards are poor. This trend highlights the role of $\alpha$ in mediating a gradual balance between local and global reward optimization, suggesting its importance in the strategic adjustment of reward focus.

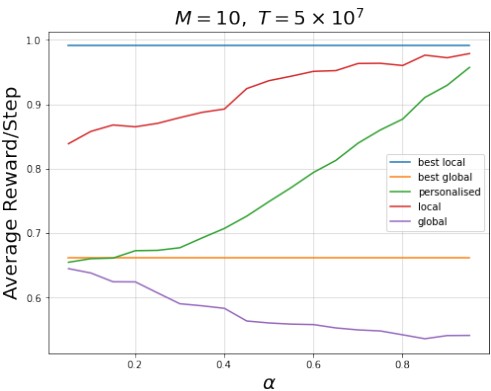

Figure 3: `PF-XAB` averaged per-step reward over varying $\alpha$. Personalised (green) adepts to $\alpha$, while local (red) and global (purple) remain fixed. Blue and yellow lines show optimal local and global values.

## 5 Conclusion

In this article we have introduced a new model for personalised, federated, bandit learning on continuous action spaces, and proposed an effective solution method utilising hierarchical partitioning, batched decision-making and optimism in the face of uncertainty. Our approach achieves a sublinear regret (evidenced both theoretically and empirically) and requires minimal communication - ensuring a good level of privacy in the federated regime.

Our method shows a near-deterministic behaviour in certain problems: we notice little variability in its regret. This is likely a result of its phased structure - the main decisions (eliminations) of the policy are made during the small number of communication rounds. If the outcomes of these decisions are identical same across replications, the expected regret will also coincide. Future work may do well do explore whether there is scope to speed up exploration through the use of a randomised policy, e.g. a variant Thompson Sampling for continuous spaces (Kandasamy et al., 2018; Grant & Leslie, 2020), or through the use of ideas from Bayesian Optimisation (Frazier, 2018) which have also proven successful in $\mathcal{X}$-armed-type problems, particularly with smooth functions.

### Acknowledgments

The authors acknowledge financial support from the Engineering and Physical Sciences Research Council (studentship ref. 2437073, https://gtr.ukri.org/projects?ref=studentship-2437073).

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
