## A    Supplementary Proofs

### A.1    Proof of Lemma 3.3

*Proof.* Let $N_{(h,i)}(p)$ represent the total count of samples collected from the node $\mathcal{P}_{(h,i)}$ during phase $p$. This total can be broken down into two components: $N^g_{(h,i)}(p) = \lceil (1-\alpha)f(p) \rceil$, the number of samples acquired during the global exploration sub-phase, and $N^l_{(h,i)}(p) = \lceil M\alpha f(p) \rceil$, the number of samples gathered during the local exploration sub-phase.

The empirical estimate $\bar{\mu}'_{(h,i),m}(p)$ can be decomposed into two parts, namely local estimates and a global estimate, such that the proxy variance satisfies

$$
\begin{aligned}
&\tilde{\sigma}^2\left(\bar{\mu}'_{(h,i),m}(p)\right) \\
&= \left(\alpha + \frac{1-\alpha}{M}\right)^2 \frac{1}{4N_{(h,i)}(p)} + \left(\frac{1-\alpha}{M}\right)^2 \sum_{n \neq m} \frac{1}{4N^g_{(h,i)}(p)} \\
&\leq \left(\frac{M\alpha + 1 - \alpha}{M}\right)^2 \frac{1}{4(M\alpha + 1 - \alpha)f(p)} + \left(\frac{1-\alpha}{M}\right)^2 \sum_{n \neq m} \frac{1}{4(1-\alpha)f(p)} \\
&= \frac{M\alpha + 1 - \alpha}{4M^2 f(p)} + \frac{(1-\alpha)(M-1)}{4M^2 f(p)} = \frac{1}{4Mf(p)},
\end{aligned}
$$

where the inequality follows by the required sampling length specified in algorithm.

$\square$

### A.2    Proof of Lemma 3.4

*Proof.* in every phase $p \in P_T$ and for each client $m \in [M]$, the Hoeffding bound can be applied to the empirical estimate of personalized objective corresponding to each node $(h,i) \in \mathcal{A}^m(p)$

$$
\mathcal{P}\left(\left|\bar{\mu}'_{(h,i),m}(p) - \mu'_{(h,i),m}\right| \geq \epsilon\right) \leq 2\exp\left(-\frac{\epsilon^2}{2\tilde{\sigma}^2\left(\bar{\mu}'_{(h,i),m}(p)\right)}\right) = 2\exp(-2\epsilon^2 Mf(p)),
$$

where it holds true because $\bar{\mu}'_{(h,i),m}(p)$ is a $\frac{1}{4Mf(p)}$-sub-Gaussian random variable, as in Lemma 3.3.

Consequently, by utilizing a union bound, the probability of event $G^c_T$, the complement of event $G_T$, can be upper bounded by

$$
\begin{aligned}
&\sum_{p \in P_T} \sum_{m=1}^M \sum_{(h,i) \in \mathcal{A}^m(p)} \mathcal{P}\left(\left|\bar{\mu}'_{(h,i),m}(p) - \mu'_{(h,i),m}\right| > B_p\right) \\
&\leq 2\sum_{p=1}^T \sum_{m=1}^M \sum_{(h,i) \in \mathcal{A}^m(p)} \exp\left(-2c^2 \log(T)\right) \\
&= 2T^{-2c^2} \sum_{p=1}^T \sum_{m=1}^M |\mathcal{A}^m(p)| \\
&\leq 2T^{-2c^2} M \sum_{p=1}^T |\mathcal{A}(p)| \leq 2M^2 T \times T^{-2c^2} \leq 2M^2 T^{-3},
\end{aligned}
\tag{5}
$$

where the second inequality is derived from the relation $\mathcal{A}^m(p) \subset \mathcal{A}(p)$. The third inequality is based on the fact that the cumulative count of active nodes across all phases is less than or equal to the total number of samples collected throughout the entire process, considering that each active node is sampled at least once. $\square$

### A.3 Proof of Lemma 3.5

*Proof.* Under the assumption of event $G_T$, the following inequality holds for every node $\mathcal{P}_{h,i} \in \mathcal{A}^m(p)$:

$$|\bar{\mu}'_{(h,i),m}(p) - \mu'_{(h,i),m}| \leq B_p. \tag{6}$$

Consequently, we have the following inequalities for the node with the node $\mathcal{P}_{h_p,i^*_{m,p}}$, at the completion of phase $p$:

$$
\begin{aligned}
\bar{\mu}'_{(h_p,i^*_{m,p}),m}(p) + B_p + \nu_1\rho^{h_p} &\geq \mu'_{(h_p,i^*_{m,p}),m} + \nu_1\rho^{h_p} \\
&\geq \mu'_m(x^*_m) \geq \mu'_{(h_p,i_p),m} \\
&\geq \bar{\mu}'_{(h_p,i_p),m}(p) - B_p,
\end{aligned}
\tag{7}
$$

where the second inequality follows from the local smoothness property of the objective function in Assumption 2.2 and $(h_p, i_p)$ is the index of node with the highest empirical estimate of personalised objective for client $m$. Thus, it can be concluded that $(h_p, i^*_{m,p})$ will not be designated for elimination for client $m$ upon the conclusion of phase $p$, or in more formal terms, $\mathcal{P}_{h_p,i^*_{m,p}} \notin \mathcal{E}^m(p)$.

$\square$

### A.4 Proof of Lemma 3.6

*Proof.* This proof consists of two parts. In the first part, we demonstrate that the mid-point of every un-eliminated node is at least $6\nu_1\rho^{h_p}$-optimal.

Under event $G_T$, for every $(h,i) \in \mathcal{A}^m(p) \setminus \mathcal{E}^m(p)$ we have the following sequence of inequalities

$$
\begin{aligned}
\mu'_{(h,i),m} + \nu_1\rho^{h_p} + 2B_p &\geq \bar{\mu}'_{(h,i),m}(p) + \nu_1\rho^h + B_p \\
&\geq \bar{\mu}'_{(h_p,i_p),m}(p) - B_p \\
&\geq \bar{\mu}'_{(h^*,i^*),m}(p) - B_p \\
&\geq \mu'_{(h^*,i^*),m} - 2B_p \\
&\geq \mu'_m(x^*_m) - \nu_1\rho^h - 2B_p,
\end{aligned}
\tag{8}
$$

where $(h^*, i^*)$ represent the node that contains the global optimum of the personalised objective. The first and fourth inequalities directly results from event $G_T$, and the second inequality follows the fact that node $(h,i)$ has not been designated for elimination at the end of phase $p$, and the third inequality follows from the definition of $(h_p, i_p)$ in Algorithm 2.

Selecting an appropriate sampling length would lead to the dominance of optimization error over statistical error, in more precise terms, this implies that $B_p \leq \nu_1\rho^h$. Therefore, by considering $f(p) = \frac{c^2 \log(T)}{M\nu_1^2\rho^{2h}}$, the latter inequality remains true, resulting in the following:

$$\mu'^*_m - \mu'_{(h,i),m} \leq 2\nu_1\rho^h + 4B_p \leq 6\nu_1\rho^h. \tag{9}$$

In the second part, we use an existing result from (Bubeck et al., 2011). Let $\mathcal{X}_{f,\epsilon} := \{x \in \mathcal{X} : f^* - f(x) \leq \epsilon\}$ denote the subset of $\mathcal{X}$ where the function $f$ is within an $\epsilon$ range of its optimal value $f^*$, representing the $\epsilon$-optimal region for $f$. According to Lemma 3 from Bubeck et al. (2011), if a node is $c\nu_1\rho^h$-optimal, then every point in the corresponding region of that node is $\max(2c, c+1)\nu_1\rho^h$-optimal.

Therefore, we can conclude that every point in the set of un-eliminated nodes at the end of phase $p$ is $12\nu_1\rho^h$-optimal, by the previous inequality. This completes the proof of the lemma. $\square$

### A.5 Regret Bound Comparison with `PF-PNE`

As we claimed in the paper, `PF-XAB` achieves the same regret bound as `PF-PNE` when operating under the assumptions of the `PF-PNE` framework. Specifically, Assumptions 3 and 4 in Li et al. (2024a) state that any $\epsilon$-near-optimal region of the global objective is uniformly $w\epsilon$-near-optimal for all local objectives for some $w \geq 1$. Personalised objectives, defined as a convex combination local and global objectives, inherit a similar near-optimality structure. Subsequently, any point in the $\epsilon$-near-optimal region of the global objective will be uniformly at least $((1-\alpha) + \alpha w)\epsilon$-near-optimal for the personalised objectives, where $\alpha$ is the weight given to the local objectives in the convex combination.

With this similarity between the near-optimal regions of personalised objectives, the regret incurred by pulling arms from the active set of other clients in the global exploration sub-phase can be bounded by this shared sub-optimality gap rather than a pessimistic upper bound of 1. Under this refinement, we get the following bound on the incurred regret during the phase $p$ under the good event $G$:

$$
\begin{aligned}
R_p^G &= R_{\text{G-expr}}^p + R_{\text{L-expr}}^p + R_{\text{expt}}^p \\
&\leq 12\nu_1\rho^{h_p-1} \times M \max_{m\in[M]} |\mathcal{A}^m(p)| \lceil (1-\alpha)f(p)\rceil + 12\nu_1\rho^{h_p-1} \times \lceil M\alpha f(p)\rceil \times \max_{m\in[M]} |\mathcal{A}^m(p)| \\
&\leq 12\nu_1\rho^{h_p-1} M f(p) \max_{m\in[M]} |\mathcal{A}^m(p)| \leq 12\nu_1\rho^{h_p-1} \times 2M \frac{2\log(T)}{M\nu_1^2\rho^{2h_p}} \times 2C\rho^{-d'h_p-1} \\
&\leq \frac{96C\log(T)}{\nu_1\rho^2}\rho^{-h_{p-1}(d'+1)}
\end{aligned}
$$

The remainder of the proof follows similarly to the main theorem.

## B Supplementary Experiments

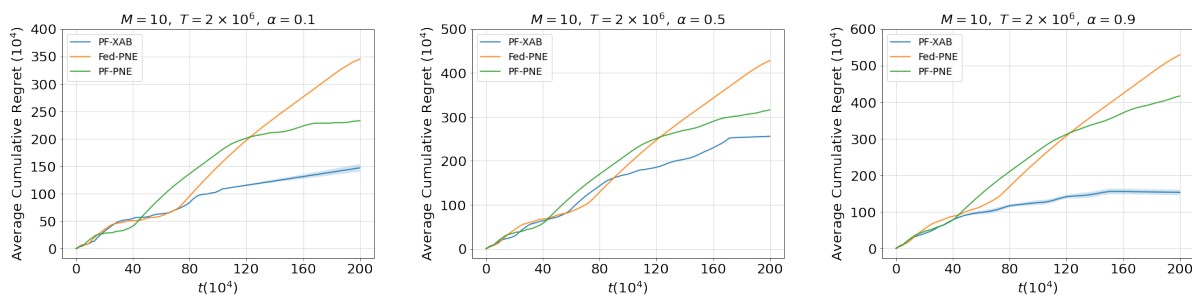

Figure 4: Cumulative regret of `PF-XAB`, `Fed-PNE`, and `PF-PNE` on DoubleSine dataset. Varying $\alpha$ (0.1 to 0.9) shows impact of personalisation. $M = 10$, $T = 2 \times 10^6$.