# OpenReview forum: "Federated $\mathcal{X}$-armed Bandit with Flexible Personalisation"
_TMLR — Accepted by TMLR_

### Review · Reviewer_XxWU · 2024-10-04

**Summary Of Contributions:**

The authors propose an $\mathcal{X}$-armed bandit-inspired federated learning (FL) framework that balances local user preferences with aggregated global knowledge. In this framework, each user has a local objective function, and the global objective is the average of these local objectives. The personalized objective for each user is then a combination of their local objective and the global objective, controlled by a personalization parameter.

The authors also introduce the PF-XAB algorithm, an $\mathcal{X}$-armed bandit-based approach designed to maximize the personalized objective through information exchange between clients and a central server. They prove that PF-XAB achieves a sub-linear regret bound and demonstrate through experiments that it outperforms existing FL $\mathcal{X}$-armed bandit algorithms.

**Audience:**

Yes

**Claims And Evidence:**

Yes

**Requested Changes:**

I found the pseudocode of the algorithms difficult to parse. It may help to break down each phase (e.g., local exploration, exploitation, etc.) in the text, with a few explanatory sentences that reference the relevant variables/statements in the pseudocode.

## Questions:
1. What does capital $P$ in the pseudocode represent?
2. It would be helpful to include a discussion of the algorithm’s space and time complexities for both the client and server sides. For example, the number of nodes in the tree appears to grow exponentially in $H$, and thus polynomial in $T$. Is that correct?

## Minor comments:
- There is a missing space after the period in the phrase "remains partially unobservable to individual clients.", which occurs just before Section 2.2.
- There’s a typo in the sentence "It **broadcast** active nodes" at the start of Section 3.1. (Should be broadcasts).

**Strengths And Weaknesses:**

## Strengths
1. The overall writing of the paper, aside from the pseudocode (which I address in requested changes), is clear and easy to follow.
2. The $\mathcal{X}$-armed bandit (XAB)-based federated learning (FL) framework with a personalization parameter makes sense, and the proposed PF-XAB algorithm fits naturally within this framework and is intuitive.
3. The authors provide theoretical guarantees for the regret of their algorithm, offering a proof (while not novel) that shows sub-linear regret.

## Weaknesses and questions
1. From what I understood, the clients and server are working asynchronously. If some clients are faster at sampling their local objectives than others, they will be forced to greedily exploit the best arm (i.e., $\left(h_p, i_p\right)$) while waiting for slower clients to finish. How is this scenario accounted for in the regret analysis? Couldn’t the faster clients end up playing suboptimal arms repeatedly due to delays from slower clients? Or is there an implicit assumption that all clients operate at the same "speed"?

2. **Experiment section**:
    1. In the first plot of Figure 1, it appears that if the simulation ran longer, PF-XAB’s regret might eventually exceed that of PF-PNE. Is this expected behavior, and if so, why?
    2. In all the plots from Figure 1, there is a noticeable bump in PF-XAB’s regret around $t=40 \times 10^4$, $30 \times 10^4$, and $20 \times 10^4$ respectively. Could the authors clarify the reason behind these bumps?
    3. In the real-world classification scenario, instead of showing cumulative regret, it would be more informative to see the average AUC-ROC score for each algorithm. Could the authors provide such results?

3. The global objective is currently defined as a simple average of local objectives, which might be restrictive in practice. Could a weighted average or other monotone-increasing functions (with smoothness assumptions) be used instead?

4. Could the authors clarify how their algorithm addresses privacy concerns? The authors mention that the server receives $\bar{\mu}_{(h, i), m}$ from each client $m$ for every node $(h, i)$ in every round. Given this, wouldn’t the server have enough information to reconstruct the rewards obtained by individual clients at each step?

5. The paper would benefit from a concrete example to ground the theoretical concepts, such as in the smart home or healthcare domains mentioned in the introduction. Could the authors provide a detailed case where they define the context space $\mathcal{X}$, the local objectives $\mu_m$ for each client, and explain what the personalized objective would represent in that scenario?

6. The algorithm needs knowledge of the smoothness parameters. A natural question is how robust is the algorithm to parameter mismatches? Specifically, if parameters related to local smoothness such as $\nu_1$ and $\rho$, and thus $H$ are not precisely known for a given problem, how does this affect the performance of the algorithm?

---

> ### Author Response · Authors · 2024-11-07
>
> We would like to thank the reviewer for their insightful comments and suggestions. We have carefully considered each point and addressed them as follows:
>
> 1. **Asynchronous Client Sampling and Regret Analysis**
>
>    Thank you for highlighting this point. In our regret analysis, equation (2) section 3.2, we bound the term $\mathbb{E}^G[R^p_{\text{L-expr}} + R^p_{\text{expt}}]$ by $\max_{m \in [M]} |\mathcal{A}^m(p)|$, which accounts for the impact of slower clients on the regret bound. This constraint on the analysis aligns with the necessity to wait for slower clients to synchronise estimations across nodes. Although enabling faster clients to proceed independently might seem advantageous, initial trials indicated that this incurs substantial regret, as explored regions may be suboptimal with large gaps relative to the personalised objectives. Therefore, this current approach balances exploration with controlled regret.
>
> 2. **Clarifications on Experimental Results**
>
>    We would like to thank the reviewer for their observations regarding the experimental results. We have addressed each question as follows:
>
>    - **Noticeable Bumps in PF-XAB's Regret around Specific Points**
>
>      The bumps observed around $t = 40 \times 10^4$, $30 \times 10^4$, and $20 \times 10^4$ in the regret plots for PF-XAB correspond to the global exploration subphase of the algorithm. During the earlier phases, exploration of regions that are active for some clients but not well-aligned with others can incur significant regret, as clients may explore areas that are suboptimal for their personalized objectives. This phenomenon causes the visible bumps in regret as the algorithm allocates time to exploring diverse regions. However, as the algorithm progresses and gradually eliminates suboptimal regions, it shifts towards local exploration and exploitation, thereby reducing the impact of these initial exploration costs. Over time, this balance neutralizes the initial bumps, leading to a more stable regret trajectory.
>
>    - **Request for Average AUC-ROC Score Instead of Cumulative Regret**
>
>      In our experiments, we calculated regret with respect to a hypothetical perfect model with an AUC-ROC of 1. This approach provides a standardized benchmark, enabling us to measure how closely each algorithm approaches optimal classification performance. Given this setup, the cumulative regret plots effectively reflect the same information as the average AUC-ROC scores.
>
> 3. **Alternative Federated Objective Functions**
>
>    We appreciate the suggestion to consider weighted averages or other monotone-increasing functions. While the uniform average aligns with the dominant federated objective framework ([1]), weighted averaging could be valuable for personalisation, as suggested in ([2]). Our proof techniques will not work in this setting without significant adaptation. The reason is that we make strong use of the fact that the uniform average over clients results in a closely synchronised behaviour for scenarios where clients are similar (homogeneous) and converge faster.
>
> 4. **Privacy Considerations**
>
>    The reviewer is correct in that we have not formally considered privacy concerns. Note however that clients share the average rewards with the server, they do not share the individual reward values ($r_{m,(h,i),t}$) obtained at each pull, and so a degree of privacy is automatic. Privacy could be formally added to the model by introducing uniform sampling from a node to select arms, and (as in [3]) adding Laplacian noise to the reported averages. We have added comments to this effect in Section 3.1, but have decided not to increase the length of the article by introducing these formally within our model.

---

> ### Author Response · Authors · 2024-11-07
>
> 5. **Concrete Example for Theoretical Concepts**
>
>    Following your suggestion, we have included a formal example in the Introduction on page 3, specifically within a clinical trials context, to clarify the theoretical concepts and the role of personalised objectives. We hope this example provides a tangible grounding for our model.
>
> 6. **Robustness to Smoothness Parameter Mismatches**
>
>    The smoothness parameters are indeed crucial as they directly influence the confidence bounds $B_p$ and the sampling lengths. Our analysis shows that if we take conservative estimates of these parameters then it will achieve regret commensurate with the conservative estimates. On the other hand, if we overestimate the smoothness then we cannot bound the regret at all. We have added a comment to this effect in Section 3.2.
>
> 7. **Clarifications and Revisions to the Pseudocode**
>
>    We appreciate the reviewer’s feedback on the pseudocode clarity. To improve readability, we have made changes to both the pseudocode and its explanation in Section 3.1. Additionally, we corrected a typographical error: the variable $P$ in the pseudocode was intended to be lowercase $p$ and this has now been fixed.
>
> Thank you again for your valuable feedback. We hope our responses and revisions address your concerns.
>
> **References:**
>
> 1. Zhang, Chen, et al. "A survey on federated learning." *Knowledge-Based Systems* 216 (2021): 106775.
>
> 2. Beaussart, Martin, et al. "Waffle: Weighted averaging for personalized federated learning." *arXiv preprint* arXiv:2110.06978 (2021).
>
> 3. Sajed, Touqir, and Or Sheffet. "An optimal private stochastic-mab algorithm based on optimal private stopping rule." *International Conference on Machine Learning.* PMLR, 2019.

---

### Review · Reviewer_Gsth · 2024-10-07

**Summary Of Contributions:**

- This work extends the previous study on federated bandits with personalization to the setup of $\chi$-armed bandits.

- To achieve flexible personalization, an algorithm termed PF-XAB is proposed, which is a phase-elimination-style algorithm with explorations designed to match the desired personalization level.

- Theoretical analyses are provided for PF-XAB, demonstrating its efficiency, which are further corroborated with experimental results.

**Audience:**

Yes

**Claims And Evidence:**

Yes

**Requested Changes:**

- For the first point in weakness, I believe it is necessary to have a more in-depth illustration of the contributions of this work, rather than extending PF-MAB to PF-XAB. The technical difficulties and novelties of handing PF-XAB should be highlighted, besides the seemingly standard algorithmic framework from PF-MAB.

- I would encourage a more in-depth illustration of the theoretical results.
  -  For example, the instance-dependent bound of Shi et al. 2021 seems to have a complete discussion on the impact of the personalization level $\alpha$ which defines the suboptimality gaps; however in Theorem 3.8 seems to have no dependency on $\alpha$, making it less interpretable.
  - As I am not an expert in $\chi$-armed bandits, I would look forward to seeing comparisons with the regret scale therein (e.g., for MAB, it is typically the form of $\log(T)/\Delta$ or $\sqrt{T}$). The current comparison with Li et al. 2024a should also be expanded to include a full context on the more restrictive assumptions therein and why the proposed PF-XAB can achieve a similar order bound.

**Strengths And Weaknesses:**

Strengths:
- This work is a valid extension of PF-MAB to $\chi$-armed bandits.

- The study is complete with algorithm designs, theoretical analysis and experimental results.

- The overall presentation is satisfying and has explained the problem/algorithm/result clearly.

Weakness:
- The main issue in my mind is that the adopted techniques to handle personalization seems to be exactly from Shi et al. 2021, i.e., the designed exploration lengths, the phased-elimination-algorithm, etc. Then, I believe the main challenge is from the $\chi$-armed bandits, however, the challenges and contributions are not sufficiently illustrated and highlighted.

- The result is not explained enough, especially its tightness. See the required changes.

---

> ### Author Response · Authors · 2024-11-07
>
> We would like to thank the reviewer for their valuable comments and suggestions. We have carefully addressed each point as follows:
>
> 1. **Contributions and Technical Novelties of PF-XAB**
>
>    Thank you for emphasising the importance of highlighting the distinct contributions and technical challenges of our work. We have expanded Section 1.1 to clarify the novel aspects of PF-XAB and to illustrate the technical difficulties unique to this $\mathcal{X}$-armed problem, beyond those in the MAB setting.
>
> 2. **Impact of Personalisation Level $\alpha$ in Theorem 3.8**
>
>    We appreciate your observation regarding the interpretability of the personalisation level $\alpha$. In response, we have added Remark 3.11.
>
> 3. **Regret Scale Comparisons and Discussion**
>
>    Thank you for this suggestion. We have expanded the discussion in Section 3.2 to provide a more comprehensive comparison of the regret scales relevant to $\mathcal{X}$-armed bandits.
>
> Thank you again for your valuable feedback. We hope our responses and revisions address your concerns.

---

### Review · Reviewer_TZgT · 2024-10-09

**Summary Of Contributions:**

The authors propose a personalised, federated learning approach building on the $\mathcal{X}$-armed bandit literature that can solve for convex combinations between local and global objectives. This approach generalises the work of [Li et al 2024a,b] as it can be applied for flexible levels of personnalisation, while relying on weaker assumptions about the homogeneity of the data. The proposed algorithm is analysed theoretically and shows good performance empirically.

**Audience:**

Yes

**Broader Impact Concerns:**

I do not find any ethical concerns about this work.

**Claims And Evidence:**

No

**Requested Changes:**

The changes that can improve the paper are answers for the weaknesses discussed above:
- Discussion of the proof techniques used and their difference with previous ones in [Li et al 2024a,b].
- Discussion of the regret rate, impact of $d$, value of $d$ in simple problems, ways to control it.
- Making the baselines work for their respective settings.
- Adding more experiments, quantify the impact of $d$ and heterogeneity on all methods.
- Correcting the small typo.


For now, I am not convinced that the experimental section supports enough the claims of the paper, however, improving it will surely make the paper ready for publication.

**Strengths And Weaknesses:**

**Strengths:**
- The paper builds on [Li et al 2024a,b] and proposes an algorithm that can deal with different levels of personnalisation (controlled by the parameter $\alpha$) while working for heterogeneous settings (requires less assumptions).
- The proposed algorithm was theoretically analysed, enjoys a sub-linear regret and is simple to run.
- The algorithm was tested on some settings and shows good performance.
- The writing is good and the paper is easy to follow.

**Weaknesses:**
- The paper can benefit from better positioning in the literature, especially in comparison to [Li et al 2024a,b]. The contribution can seem very incremental, and discussions about what makes the convex combination problem more difficult should be included. For example, the differences in the technical solutions are not discussed: Is your algorithm an adaptation of the previous ones [Li et al 2024a,b]? Where does the difficulty reside? These insights will make the paper richer.

- The proposed algorithm was analysed theoretically and presents a sub-linear regret that deteriorates the rate of previous approaches. This is not an issue as it is supposed to solve a more general problem. This deterioration seems to come from the drop of the homogeneity assumptions and it would be nice to have a discussion on this, even a proof in the appendix. The rate also depends on the the constant $d$ that is not much discussed. What is its value in simple problems? is $d$ very big to the point of getting linear regret? Is there a way to control its value?

- The experimental section shows good performance for the proposed methods, but does not make much sense for the baselines. If we restrict ourselves to the first experiment, we should expect that $\alpha \rightarrow 0$ would make Fed-PNE work better than PF-PNE as it is tailored to solve the global objective, and $\alpha \rightarrow 1$ will make the newly proposed method and PF-PNE comparable. This is not the case as Fed_PNE is always performing poorly in $\alpha \rightarrow 0$ and PF-PNE shows **worse than linear regret** in $\alpha \rightarrow 1$, which are favorable cases for the algorithms. Is there any explanation to this?

- The experimental section would benefit from more experiments. It would be also nice to have different settings with different constants $d$ and different levels of heterogeneity to understand the differences between the algorithms better and quantify their impact on the regret.

- Small typo/ unclear definition: I do not understand the notation with which you define $\mathcal{P}_{h, i}$, are there missing indices in the union?

---

> ### Author Response · Authors · 2024-11-07
>
> We would like to thank the reviewer for their constructive comments and suggestions. We have carefully addressed each point as follows:
>
> 1. **Positioning in the Literature and Technical Challenges**
>
>    We appreciate the suggestion to better position our work within the existing literature and clarify the unique challenges of the problem. In our revision, we have expanded Sections 1.1 and 3.2 to discuss the technical distinctions of PF-XAB.  PF-XAB, Fed-PNE, and PF-PNE share the core strategies of hierarchical partitioning and phased elimination. These strategies are essential for efficiently exploring and exploiting the continuous action space inherent in X-armed bandit problems. However, PF-XAB presents a unique challenge by optimising personalised objectives, where each client's objective is a convex combination of its local objective and the global (average) objective. This formulation allows clients to benefit from the shared knowledge while respecting individual preferences, but it also  requires the algorithm to balance potentially conflicting goals of optimising both local and global objectives. We have elaborated on these challenges in the revised sections.
>
> 2. **Regret Rate and Homogeneity Assumptions**
>
>    Thank you for this observation. We have now included a sketch proof in the Appendix to indicate that PF-XAB performs as well as PF-PNE in the homogeneous cases, along with a comment in Section 3.2 discussing this fact.
>
> 3. **Impact of the Constant $d$ on Regret Rate**
>
>    We appreciate your interest in understanding the role of the near-optimality dimension $d$ in determining the regret rate. We have expanded Section 3.2 and added Remark 3.8 to provide further insights.
>
> 4. **Experimental Results for Baselines in Different $\alpha$ Settings**
>
>    Thank you for this observation. In Figure 1, the results show that even at the extreme values of $\alpha = 0.1$ and $\alpha = 0.9$, personalisation effects still influence the performance. For $\alpha = 0.1$, PF-PNE and PF-XAB outperform Fed-PNE because a 10% weight on the personalisation term allows these methods to leverage individual preferences, which Fed-PNE does not address as effectively. For $\alpha = 0.9$, the data heterogeneity favors PF-XAB over PF-PNE, suggesting that methods like PF-XAB, designed to handle high heterogeneity, gain a performance advantage.
>
> 5. **Additional Experimental Settings**
>
>    Thank you for suggesting additional experiments with varying constants $d$ and different levels of heterogeneity. We appreciate the opportunity to expand on this aspect. From the definition of the stopping depth criterion $H$, we know that increasing $d$ causes a decrease in $H$, which effectively leads to earlier halting of global exploration as $d$ increases. Our experiment confirmed this, showing that higher values of $d$ lead to a shorter exploration phase. Consequently, the regret profiles for different values of $d$ can appear similar if they yield equivalent stopping depths, as the exploration phases conclude at the same point. However, when the stopping depth differs, regret differences only become evident after the exploration phase ends. This is because the algorithms continue to explore until reaching their respective stopping depths. Once exploration has halted, differences in regret become noticeable, as higher values of $d$ lead to earlier stopping, thereby affecting cumulative regret in subsequent phases.
>
> 6. **Notation for $\mathcal{P}_{h,i}$**
>
>    We apologize for the confusion regarding the notation. $\mathcal{P}_{h,i}$ refers to a specific node within the hierarchical partition. The subscript $h$ indicates the depth of the node in the hierarchy. A depth of 0 refers to the root node, which encompasses the entire space $\mathcal{X}$. As $h$ increases, the nodes represent finer subdivisions of the search space. The subscript $i$ is the index of the node at a given depth. At each depth, multiple nodes with different indices exist to cover the entire space (where $0 < i < 2^h$).
>
> Thank you again for your valuable feedback. We hope our responses and revisions address your concerns.

---

### Review · Reviewer_SYhh · 2024-10-16

**Summary Of Contributions:**

This paper studies federated X-armed bandits, where a set of clients want to optimise a convex combination between their local and global objectives controlled by a “personalisation parameter” $\alpha$. The paper proposes an elimination-based algorithm PF-XAB which enjoys a sublinear regret $\tilde{O}(T^\frac{d’+2}{d’ + 3})$ where $d’$ is a quantity related to the near-optimality dimension.

**Audience:**

Yes

**Claims And Evidence:**

Yes

**Requested Changes:**

The requested changes are answers to the weakness above. Specifically:
- A paragraph on the challenges of this setting in comparison to [Shi et al. (2021)] and [Li et al. (2024a)]. Specifically, [Shi et al. (2021)] studies the same mixed objective, but for finite-armed bandits. Does the hardness of your setting reduce to the hardness of going from finite-armed to X-armed bandits in [Shi et al. (2021)]?
- A communication cost analysis of the PF-XAB algorithm, with the number of communication rounds explicit with respect to the parameters of PF-XAB
- A proof that PF-XAB retrieves the same regret upper bound as [Li et al. (2024a)] with the additional two assumptions.
- A discussion on the parameter $d’$ in the regret upper bound. Both a theoretical/experimental intuition about this parameter may be valuable. Also, comparing how this quantity relates to similar quantity appearing in regret upper bounds of X-armed bandits ([Li et al. (2024a)], [Bubeck et al. (2011)]) is interesting.
- A discussion on the effect of $\alpha$ on the regret upper bound.
- Even if the regret lower bound for this specific setting seems to be left as an open problem, a discussion on the lower bounds in similar settings ([Bubeck et al. (2011)], [Shi et al. (2021)] and [Li et al. (2024b)]) and how they relate to your upper bound may provide valuable insights on the hardness of this setting.

**Strengths And Weaknesses:**

Strengths:
- The paper is well-written and the setting is well-motivated
- The paper proposes a theoretical analysis of the regret of the proposed PF-XAB algorithm
- The paper validates experimentally the superiority of PF-XAB in the personalised setting (i.e. $\alpha \neq 0$), compared to algorithms either tailored for local or global objectives under both synthetic and real-world datasets.


Weakness:
- The challenges/specific hardness of this setting compared to that of [Shi et al. (2021)] and [Li et al. (2024a)] are not fully addressed.
- The paper affirms that PF-XAB maintains a “logarithmic communication cost”. However, a rigourous analysis of the communication cost of PF-XAB is missing in the paper.
- The paper claims at the end of Section 3 that the regret of PF-XAB would retrieve the same regret upper bound of [Li et al. (2024a)] if the two additional assumptions from [Li et al. (2024a)] were added. However, this claim is not obvious, and no rigorous/sketch proof is provided.
- The regret upper bound mainly depends on the parameter $d’$, which seems hard to parse. Specifically, how does this quantity relate to $d_m$ in [Li et al. (2024a)] and to $d’$ in [Bubeck et al. (2011)]?
- The regret upper bound hides the dependence on the “personalisation parameter” $\alpha$, which seems to be a critical quantity.
- No regret lower bounds are provided for this setting.

---

> ### Author Response · Authors · 2024-11-07
>
> We would like to thank the reviewer for their detailed feedback and suggestions. We have addressed each point as follows:
>
> 1. **Challenges Compared to [Shi et al. (2021)] and [Li et al. (2024a)]**
>
>    Thank you for suggesting a deeper comparison with the challenges in [Shi et al. (2021)] and [Li et al. (2024a)]. We have added discussions in Sections 1.1, 2.1, and 3.2 to clarify the unique challenges of our setting. While [Shi et al. (2021)] addresses mixed objectives in finite-armed bandits, extending this to an X-armed bandit setting introduces added complexity due to the continuous action space, which requires a distinct approach by employing a hierarchical partitioning and phased elimination.
>
> 2. **Communication Cost Analysis**
>
>    We appreciate this observation regarding the communication cost. To address this, we have added a statement on the communication cost of PF-XAB as Remark 3.10, where we explicitly discuss the number of communication rounds with respect to the parameters of PF-XAB.
>
> 3. **Regret Bound Comparison with [Li et al. (2024a)]**
>
>    Thank you for pointing out the need for additional clarification on this claim. We have included a sketch proof in the Appendix, as well as a comment in Section 3.2, to show that PF-XAB achieves the same regret upper bound as [Li et al. (2024a)] when the additional assumptions from their work are applied.
>
> 4. **Explanation of the Parameter $d'$ and its Relationship to $d_m$ in [Li et al. (2024a)] and $d'$ in [Bubeck et al. (2011)]**
>
>    In our setting, $d'$ represents the maximum near-optimality dimension across all personalised objectives. Our regret bound indicates that the client with the most complex local landscape (i.e., with the largest near-optimality dimension) influences the overall regret. In both Fed-PNE [Li et al. (2024a)] and HOO [Bubeck et al. (2011)], $d'$ corresponds to the near-optimality dimension of the global objective and the objective function itself, respectively. In PF-PNE, the regret bound depends on both the near-optimality dimension of the global objective and the maximum near-optimality dimension among local objectives, resembling a similar dependency pattern to our approach.
>
> 5. **Effect of Personalisation Parameter $\alpha$ on the Regret Upper Bound**
>
>    Thank you for raising this point. We have discussed the impact of the personalisation parameter $\alpha$ on the regret bound in Remark 3.11.
>
> Thank you again for your valuable feedback. We hope our responses and revisions address your concerns.

---

### Decision · Action_Editor_Rs4o · 2024-11-22

**Recommendation:** Accept as is

**Comment:**

As explained above, reviewers all agree that the paper meets both the "claims and evidence" and "audience" criteria for acceptance in TMLR. I agree with the reviewers' assessment.

**Audience:**

Reviewers agree that the $\mathcal{X}$-armed-bandit inspired federated learning setting is intuitive and well-motivated. Reviewers also point out that the paper is well-written, with a proposed algorithm demonstrating good experimental performance. As such, the paper will be of interest to some parts of the TMLR audience/community.

**Claims And Evidence:**

The authors have addressed reviewers' concerns about experiments (to support the claims in the paper), and have also added additional results and contexts comparing against prior related work. Reviewers unanimously agree that the "claims and evidence" acceptance criterion for TMLR is satisfied.